# Facile Fabrication of a Novel Au/Phosphorus-Doped g-C$_3$N$_4$ Photocatalyst with Excellent Visible Light Photocatalytic Activity

**Hao Li [1,2], Nan Zhang [1,2], Fei Zhao [1,2], Tongyao Liu [1,2] and Yuhua Wang [1,2,*]**

[1] Department of Materials Science, School of Physical Science and Technology, Lanzhou University, Lanzhou 730000, China; lihao3218@163.com (H.L.); zhangn2018@lzu.edu.cn (N.Z.); zhaof2016@lzu.edu.cn (F.Z.); liuty2011@lzu.edu.cn (T.L.)

[2] National and Local Joint Engineering Laboratory for Optical Conversion Materials and Technology, School of Physical Science and Technology, Lanzhou University, Lanzhou 730000, China

* Correspondence: wyh@lzu.edu.cn; Tel.: +86-931-8912772; Fax: +86-931-8913554

**Abstract:** The intrinsic disadvantages of pristine graphitic carbon nitride (g-C$_3$N$_4$) significantly restrict its applications in photocatalysis field. Hence, we have demonstrated facile thermal copolymerization and in situ photodeposition methods to fabricate a novel Au/phosphorus-doped g-C$_3$N$_4$ (Au/P-g-C$_3$N$_4$) photocatalyst. The results showed that phosphorus was doped into the structure of g-C$_3$N$_4$ and that the surface deposition of gold was successfully accomplished. The H$_2$ generation rate of the optimal Au/P-g-C$_3$N$_4$ is 8.4 times compared with the pristine g-C$_3$N$_4$ under visible light irradiation. The enhancement of photocatalytic activity is due to the synergic effect between gold induced surface plasmon resonance and the modified structural and electronic properties of the g-C$_3$N$_4$ induced by the phosphorus dopant.

**Keywords:** g-C$_3$N$_4$; surface plasmon resonance effect; phosphorus doping; photocatalytic activity

---

## 1. Introduction

Photocatalytic hydrogen (H$_2$) generation using solar energy is the most promising strategy to solve the energy crisis [1–4]. To date, various semiconductors have been used for photocatalytic H$_2$ generation, such as ZnO, Ag$_2$S and BiVO$_4$. They are mainly inorganic semiconductors with poor photocatalytic activity [5–7]. Hence, photocatalysts with excellent activity for H$_2$ generation using solar energy are still unavailable, which remains a significant challenge [8].

Graphitic carbon nitride, g-C$_3$N$_4$, is a non-metal polymer semiconductor that has shown much promise as a visible light photocatalyst for a range of applications from H$_2$ generation to organic pollutant digestion [9]. It has a band gap of about 2.6 eV (urea thermal polymerization) [10]. g-C$_3$N$_4$ is an organic semiconductor showing the advantages of an environmentally friendly and sustainable material compared with other reported photocatalysts. Meanwhile, it consists of easily acquired elements from our planet (carbon and nitrogen) [11]. The previous research has clearly shown that the conjugated aromatic π-stacking g-C$_3$N$_4$ photocatalyst owns satisfactory chemical and thermal stability as well as good electronic properties. As a consequence, g-C$_3$N$_4$ becomes a promising material in the photocatalysis field [11,12]. However, some important problems still limit the application of g-C$_3$N$_4$ in the photocatalytic field, such as, the narrow absorption region, the poor surface area and low separation rate of the photogenerated electron-hole [13]. Thus, many strategies such as nonmetal or metal doping, coupling with noble metal materials (Pt, Au, Ag) or carbon materials, introducing vacancy have been employed [14–17].

In the above methods, doping with nonmetal atoms and coupling Au nanoparticles have been considered as an efficient strategy. Recently, Cheng et al. prepared Sulfur doped g-$C_3N_4$ via treated raw g-$C_3N_4$ at high temperature in $H_2S$ atmosphere. However, it could pollute the surrounding environment and S-doped g-$C_3N_4$ exhibited unsatisfactory photocatalytic activity [18]. Zhang et al. employed ionic liquid and dicyandiamide as precursors to prepare phosphorus doped g-$C_3N_4$. The visible light utilization ability was obviously improved after phosphorus doping. Nevertheless, it is not a facile method because of the high cost of ionic liquid and complicated synthesis [19]. Furthermore, Qian et al. reported on Au nanoparticles modified on the surface of g-$C_3N_4$ through a complex method and surveyed the photocatalytic activity for $H_2$ generation [20]. Parida et al. dispersed Au nanoparticles on g-$C_3N_4$ to transfer photogenerated electrons and to improve the light absorption [11]. The photocatalytic activity of g-$C_3N_4$ could be partly improved with the existence of Au nanoparticles. However, the deposition and combination of Au nanoparticles on the surface of g-$C_3N_4$ were not good. In general, the early studies did not reflect the benefit of the synergic effect between doping and surface plasmon resonance. As a consequence, the reinforcement of photocatalytic performance in the above works was not marked.

Hence, we fabricated Au/phosphorus-doped g-$C_3N_4$ (Au/P-g-$C_3N_4$) photocatalyst by facile thermal copolymerization and in situ photodeposition methods. A series of analyses evidenced the good electronic structures and texture of Au/P-g-$C_3N_4$. Subsequently, the reasons for enhanced photocatalytic activity are discussed and a possible mechanism is proposed based on our experimental results.

## 2. Results and Discussion

The crystal structure of as-prepared g-$C_3N_4$, P-g-$C_3N_4$, 3% Au/g-$C_3N_4$ and different Au/P-g-$C_3N_4$ samples were analyzed by XRD. As shown in Figure 1, the XRD pattern of g-$C_3N_4$ revealed two distinct reflections at 13.47 and 27.25°, which could be easily indexed that the g-$C_3N_4$ belonged to hexagonal phase (JCPDS 87-1526). The diffraction peak at 27.25° was assigned to the (002) plane of graphitic g-$C_3N_4$, which resulted from the conjugated aromatic system's stacking. Another minor peak at 13.47° was indexed for the (100) plane of g-$C_3N_4$ derived from the repeated units of melem (1,3,4,6,7,9,9b-Heptaazaphenalene-2,5,8-triamine) [21]. For P-g-$C_3N_4$, the above peaks were still maintained. This meant the phosphorus doping process did not change the structure of g-$C_3N_4$. Additionally, the peak position of the (002) plane shifted from 27.25 to 26.65°, which could probably be ascribed to the P doped into the structure of g-$C_3N_4$, indicating the interlayer stacking distance of P-g-$C_3N_4$ was increased [1]. This meant that g-$C_3N_4$ was exfoliated (larger surface area). The overall weakening intensity of the peaks concluded that P-g-$C_3N_4$ have poorer crystallinity, also resulting from the marginally doping of P which inhibited the growth of the g-$C_3N_4$ [22]. For Au nanoparticles loaded all samples, the XRD patterns revealed four separate reflections at 38.3, 44.5, 64.5, and 77.7°, corresponding to (111), (200), (220), (311) planes of metallic Au, respectively. The observed peaks were well matched with the featured reflections of Au (JCPDS 01-1174) [11]. The peaks of the above mentioned gradually increased with increasing the Au loading percentage on P-g-$C_3N_4$. The four peaks indicated the composition of Au on the P-g-$C_3N_4$, which was not discovered in the raw g-$C_3N_4$. The presence of both Au planes and P-g-$C_3N_4$ confirmed the formation of nanocomposites. Meanwhile, it could be concluded that impurities or any other phases did not exist in the composites because no significant other diffraction reflections could be observed, which showed that the introduction of Au particles and P doping did not change the crystal structure of g-$C_3N_4$.

The valence states and chemical composition of P-g-$C_3N_4$ and 3% Au/P-g-$C_3N_4$ were analyzed by XPS. Figure 2a shows the survey spectrum of P-g-$C_3N_4$. The spectrum manifested the presence of C, N, P and surface absorption O, which meant the thermal copolymerization process introduced P into g-$C_3N_4$ structure. Figure 2b shows the survey spectrum of 3% Au/P-g-$C_3N_4$, the peaks belonging to C, N, P and Au indicated the deposition of Au had been accomplished. More importantly, the P 2p signal (Figure 2c) could be fitted as two peaks located in 133.1 eV and 133.9 eV. These peaks were attributed to the P-N species, indicating P replaced C atoms in the g-$C_3N_4$ structure [23]. The high-resolution XPS

data was clear proof that P marginally doped into the structure of g-C$_3$N$_4$. Meanwhile, the two distinct peaks located in 79.8 eV and 83.6 eV (Figure 2d) originated from Au 4f$_{5/2}$ and 4f$_{7/2}$ electrons of metallic Au [24]. Therefore, it could be confirmed that the photodeposition process effectively generated metallic Au$^0$ species on the surface of P-g-C$_3$N$_4$. As shown in Table 1, the P external element content was determined to be 0.66 wt.% and Au external element content was determined to be 2.96 wt.% (close to nominal Au content) by XPS analysis for 3% Au/P-g-C$_3$N$_4$, also the corresponding C/N atomic ratio (0.74) was close to the theoretical value of raw g-C$_3$N$_4$ (0.75) [25]. These results showed that the deposition of Au and doping of phosphorus into the texture of g-C$_3$N$_4$ had been accomplished. This was also supported by XRD studies.

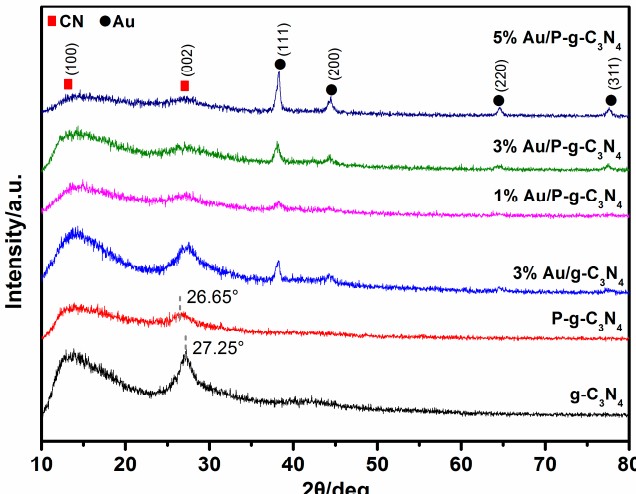

**Figure 1.** XRD spectrograms of g-C$_3$N$_4$, P-g-C$_3$N$_4$, 3% Au/g-C$_3$N$_4$ and different Au/P-g-C$_3$N$_4$ samples.

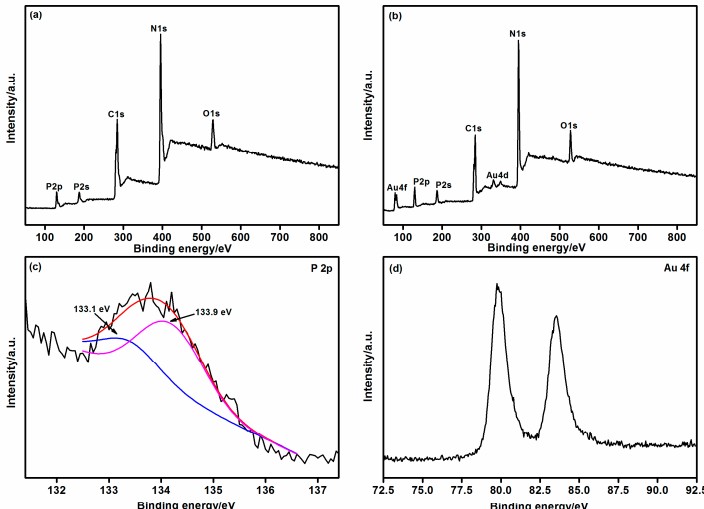

**Figure 2.** XPS survey spectra of P-g-C$_3$N$_4$ (**a**) and 3% Au/P-g-C$_3$N$_4$ (**b**), P 2p high-resolution spectrum (**c**) and Au 4f high-resolution spectrum (**d**) for 3% Au/P-g-C$_3$N$_4$.

**Table 1.** XPS data obtained for P-g-C$_3$N$_4$ and 3% Au/P-g-C$_3$N$_4$.

| Samples | Binding Energy of P 2p | Binding Energy of Au 4f$_{5/2}$ and 4f$_{7/2}$ | Surface C/N Atomic Ratio | P cont. [wt.%] | Au cont. [wt.%] |
|---|---|---|---|---|---|
| P-g-C$_3$N$_4$ | 133.7 eV | - | 0.74 | 0.67 | - |
| 3% Au/P-g-C$_3$N$_4$ | 133.8 eV | 79.8 eV and 83.6 eV | 0.74 | 0.66 | 2.96 |

The morphology and structure of 3% Au/P-g-C$_3$N$_4$ were analyzed by TEM. The 3% Au/P-g-C$_3$N$_4$ showed morphology of ultrathin nanosheets (Figure 3a). These nanosheets were soft and loose, which meant that g-C$_3$N$_4$ was exfoliated during the doping process. The nanosheets possessed a larger surface area and abundant reactive sites. Meanwhile, Au nanoparticles (black colored dots) were uniformly and adequately anchored on the surface of P-g-C$_3$N$_4$ nanosheets, which can be found in Figure 3b clearly. We could observe that a strong combination between Au and P-g-C$_3$N$_4$, ensuring the electrons transfer between Au and P-g-C$_3$N$_4$ smoothly. This strong combination also enhanced plasmonic properties of the 3% Au/P-g-C$_3$N$_4$. Figure 3c shows the HRTEM image of single Au nanoparticle. The lattice spacing of 0.203 nm belonged to the (200) lattice planes of metallic Au [24]. The composition of 3% Au/P-g-C$_3$N$_4$ was analyzed by EDS. The EDS spectrum of 3% Au/P-g-C$_3$N$_4$ showed characteristic peaks of C, N, P and Au (Figure 3d). This data was also in accordance with XRD and XPS results. Meanwhile, mean particle size and particle size distribution of all Au loaded samples are shown in Table 2. TEM images for the whole series of catalysts are shown in Figure S1. The BET surface areas of samples were calculated used nitrogen adsorption-desorption isotherms of samples (Figure S2) [26]. The BET specific surface areas of g-C$_3$N$_4$, P-g-C$_3$N$_4$ and 3% Au/P-g-C$_3$N$_4$ were 63.2, 77.5 and 77.8 m$^2$/g, respectively. The BET specific surface area of P-g-C$_3$N$_4$ was slightly larger than that of raw g-C$_3$N$_4$, which could have resulted from the P doped into the structures of g-C$_3$N$_4$, indicating that the interlayer stacking distance of P-g-C$_3$N$_4$ was increased. This meant that the process of doping was exfoliated (larger surface area). Moreover, the BET specific surface areas of P-g-C$_3$N$_4$ and 3% Au/P-g-C$_3$N$_4$ were uniform, this meant that the deposition process did not change BET surface area.

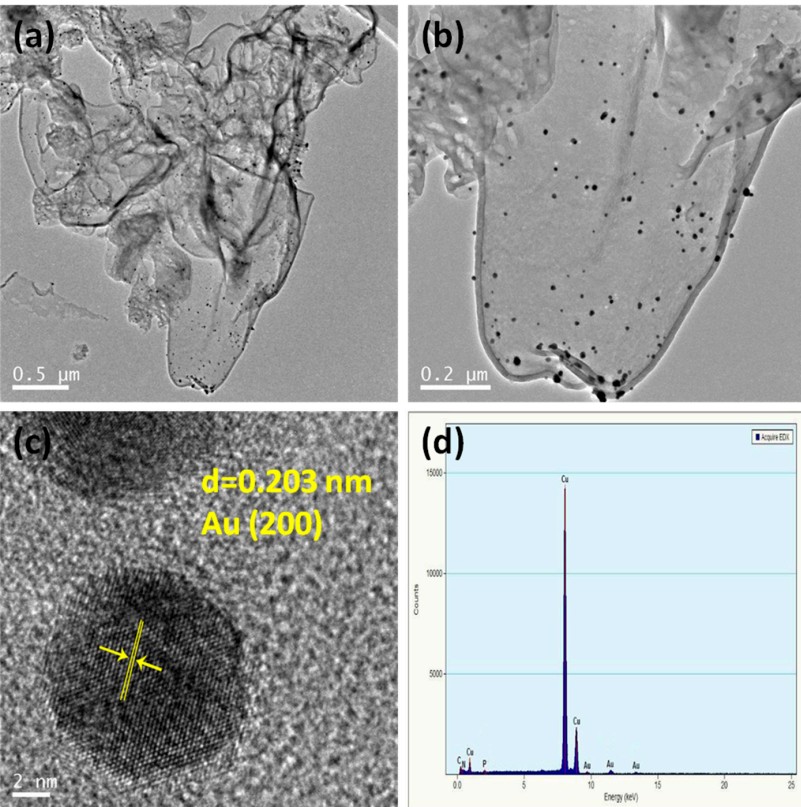

**Figure 3.** TEM image with low-magnification (**a**) of 3% Au/P-g-C$_3$N$_4$ and high-magnification (**b**) of 3% Au/P-g-C$_3$N$_4$. (**c**) Corresponding HRTEM image of a single Au nanoparticle and (**d**) EDS spectrum of 3% Au/P-g-C$_3$N$_4$.

**Table 2.** Mean particle size and particle size distribution of all Au loaded samples.

| Samples | Mean Particle Size | Particle Size Distribution |
|---|---|---|
| 3% Au/g-C$_3$N$_4$ | 24 nm | 13–56 nm |
| 1% Au/P-g-C$_3$N$_4$ | 12 nm | 8–33 nm |
| 3% Au/P-g-C$_3$N$_4$ | 22 nm | 11–54 nm |
| 5% Au/P-g-C$_3$N$_4$ | 70 nm | 20–130 nm |

UV-vis diffuse reflectance spectroscopy was used to investigate the optical absorption of the obtained samples. As shown in Figure 4, the band gap of g-C$_3$N$_4$ was 2.59 eV (inset), which exhibited an absorption edge located at 479 nm. Remarkably, the band gap of P-g-C$_3$N$_4$ was 2.47 eV, which exhibited a stronger absorption edge located at 502 nm. The reason for the absorption edge's red shift was that the P 3p state was located at the bottom of the conduction band in the g-C$_3$N$_4$ host, resulting in the decrease of band gap [27]. This means P-g-C$_3$N$_4$ have improved visible light utilization ability. Meanwhile, with the incorporation of Au, the SPR of Au nanoparticles resulted in an intense absorption in the region of 450–700 nm. An intense near-electric field could be produced due to this SPR absorption on the interface between Au and P-g-C$_3$N$_4$, which promoted the separation rate of photogenerated charge [20]. In addition, some of hot electrons generated by the SPR of Au could directly inject into the conduction band of the P-g-C$_3$N$_4$ crossing the Schottky barrier, improving the light utilization abilities of Au/P-g-C$_3$N$_4$ nanocomposites [28]. Band gap, relative conduction band and valence band of all obtained samples are shown in Table 3.

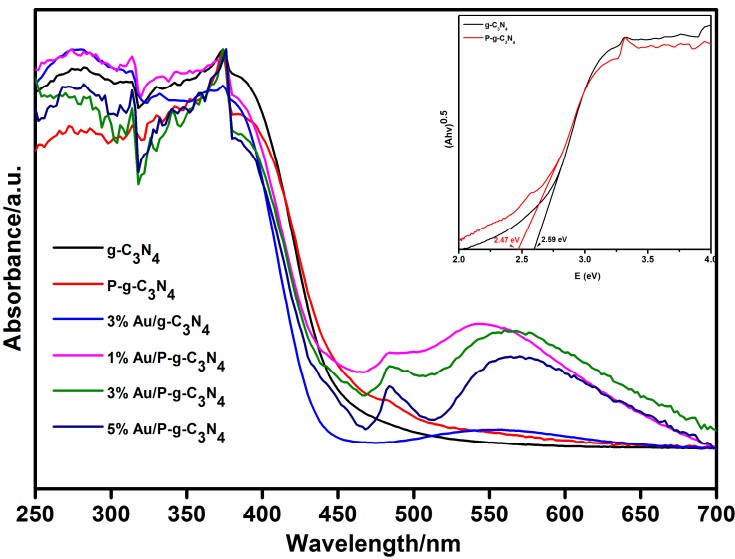

**Figure 4.** UV-visible absorption spectra of g-C$_3$N$_4$, P-g-C$_3$N$_4$, 3% Au/g-C$_3$N$_4$ and different Au/P-g-C$_3$N$_4$ samples. Corresponding band gap calculation (inset).

**Table 3.** Band gap (Eg), relative conduction band ($E_{CB}$) and valence band ($E_{VB}$) of all obtained samples.

| Samples | Eg (eV) | $E_{CB}$ | $E_{VB}$ |
|---|---|---|---|
| g-C$_3$N$_4$ | 2.59 | −1.07 | 1.52 |
| P-g-C$_3$N$_4$ | 2.47 | −1.01 | 1.46 |
| 3% Au/g-C$_3$N$_4$ | 2.58 | −1.07 | 1.51 |
| 1% Au/P-g-C$_3$N$_4$ | 2.45 | −1.00 | 1.45 |
| 3% Au/P-g-C$_3$N$_4$ | 2.47 | −1.01 | 1.46 |
| 5% Au/P-g-C$_3$N$_4$ | 2.46 | −1.01 | 1.45 |

The photocatalytic activities for H$_2$ generation of prepared g-C$_3$N$_4$, P-g-C$_3$N$_4$, 3% Au/g-C$_3$N$_4$ and different Au/P-g-C$_3$N$_4$ composites were assessed in methanol aqueous solution under visible light

irradiation. As shown in Figure 5a, the P-g-$C_3N_4$ showed higher photocatalytic activity than g-$C_3N_4$. The reason was the narrow band gap and large surface area of P doped g-$C_3N_4$. The narrow band gap and large surface area meant abundant reactive sites and improved visible light utilization ability. When g-$C_3N_4$ was coupled with Au nanoparticles, 3% Au/g-$C_3N_4$ showed enhanced photocatalytic activities compared with g-$C_3N_4$. The reason was the SPR effect of Au, which brought an advanced separation rate of photogenerated charge. The 3% Au/P-g-$C_3N_4$ sample exhibited the strongest photocatalytic activity for $H_2$ generation and the $H_2$ generation rate was 8.4 times compared with the pristine g-$C_3N_4$ under visible light irradiation. The improvement of photocatalytic activity of 3% Au/P-g-$C_3N_4$ could be ascribed to the above-mentioned reasons. These reasons could be summed up as synergic effect between gold induced surface plasmon resonance and the modified structural and electronic properties of the g-$C_3N_4$ induced by the phosphorus dopant. Furthermore, from the practical applications of 3% Au/P-g-$C_3N_4$, the photocatalytic stability of 3% Au/P-g-$C_3N_4$ sample was evaluated under the same conditions. The 3% Au/P-g-$C_3N_4$ exhibited a good catalytic stability (Figure 5b), keeping a similar level of activity after four cycles. Therefore, Au/P-g-$C_3N_4$ had a potential practical application value. Reacted 3% Au/P-g-$C_3N_4$ was analyzed using XRD, XPS and TEM. The results are presented in Figures S3 and S4. By comparison, distinct XRD diffraction peaks of fresh 3% Au/P-g-$C_3N_4$ and used 3% Au/P-g-$C_3N_4$ were almost changeless, implying that the chemical structure of 3% Au/P-g-$C_3N_4$ was maintained. As shown in Figure S4a, the P 2p signal was attributed to the P-N species, indicating P replaced C. The Au 4f signal means $Au^0$ species were dispersed on the surface of P-g-$C_3N_4$. As shown in Figure S4b, the morphology and mean particle size of 3% Au/P-g-$C_3N_4$ was not changed after the reaction. The P and Au contents of the fresh catalyst determined by ICP were 0.62 and 2.86 wt.%. The P and Au contents of the used catalyst determined by ICP were 0.61 and 2.84 wt.%. These results all implied that the chemical structure of 3% Au/P-g-$C_3N_4$ was maintained with minimal leaching of P and Au.

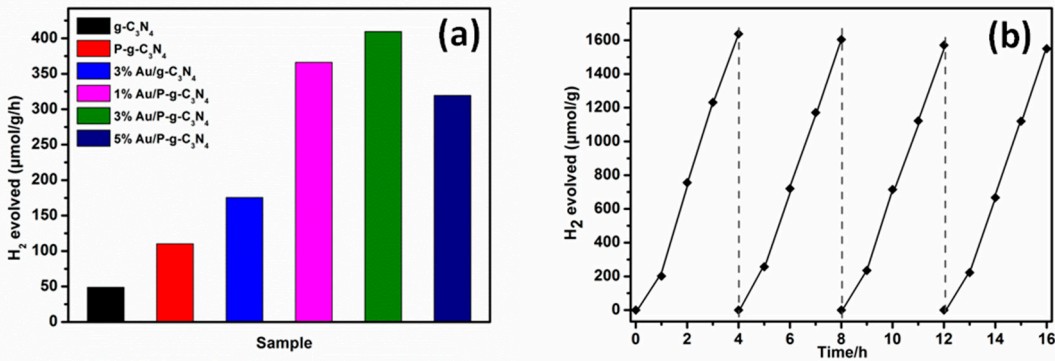

**Figure 5.** Photocatalytic activities for (**a**) $H_2$ generation rates for prepared photocatalysts under visible light illumination. (**b**) Circling runs of 3% Au/P-g-$C_3N_4$ composite for $H_2$ generation rates under visible light irradiation.

The separation efficiency of the photogenerated charge was revealed by photoluminescence analysis. Figure 6 shows the spectra of all prepared samples excited by 365 nm. The PL intensity of nanocomposites had a significant decrease compared with the raw g-$C_3N_4$ and P-g-$C_3N_4$. The weaker the PL peak's intensity appeared, meant the lower the recombination rate of the photogenerated charge possessed [29].

For the purpose of further confirmation of the efficiency of the carrier separation and transfer, the transient photocurrent response data was collected by an electrochemical workstation. The photocurrent was measured under $\lambda \geq 420$ nm. The photoexcited electrons contributed to the photocurrent generation in our experiment. As shown in Figure 7, the g-$C_3N_4$ exhibited poor photocurrent density. The reason was limited visible light utilization and the high recombination rate of charge. The 3% Au/P-g-$C_3N_4$ sample had the strongest photocurrent density, the photocurrent

density attained as high as 15 μA/cm$^2$. The excellent photocurrent density could be attributed to the enhanced light utilization ability and high efficiency of carrier separation and transfer, which resulted from the SPR effect of Au and P doping. The on-off cycles of photocurrent were quick-response and reversible, which meant the current was truly photogenerated [30].

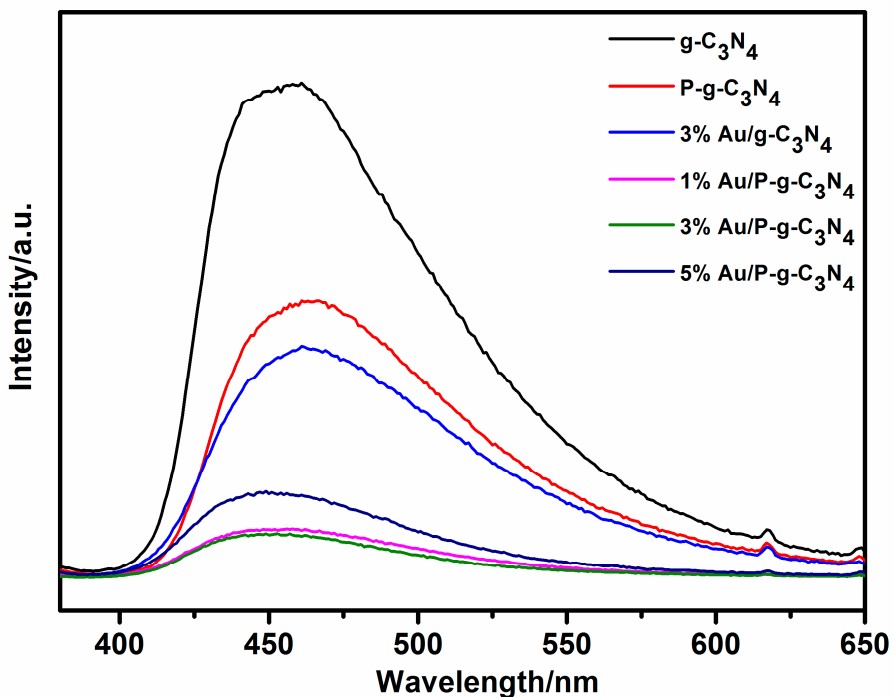

**Figure 6.** PL spectra of g-C$_3$N$_4$, P-g-C$_3$N$_4$, 3% Au/g-C$_3$N$_4$ and different Au/P-g-C$_3$N$_4$ composites.

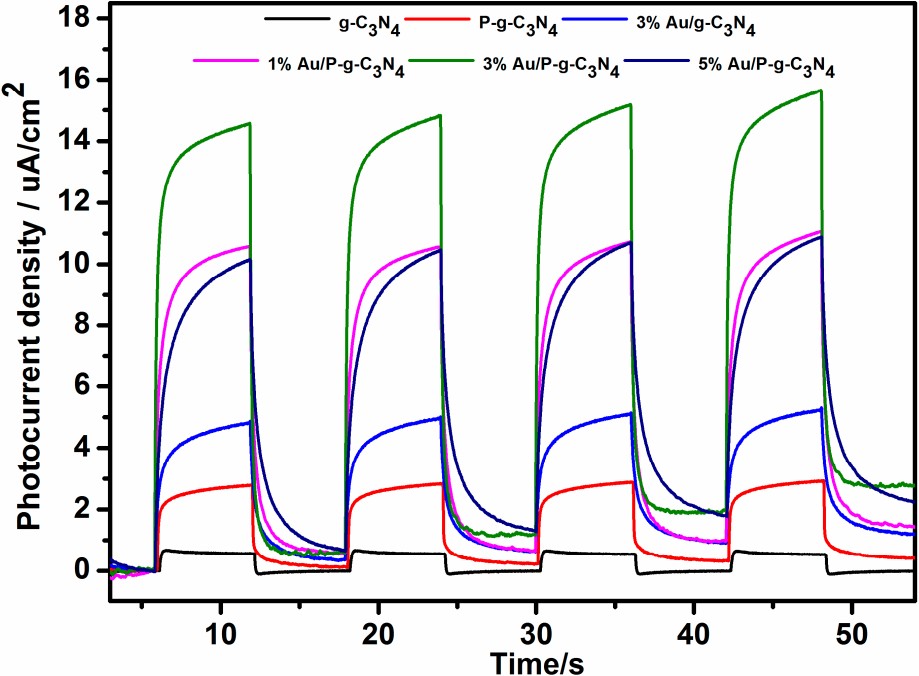

**Figure 7.** Transient photocurrent density of obtained samples.

Based on the above discussion, the mechanical illustration of 3% Au/P-g-C$_3$N$_4$ in the photocatalytic reaction is shown in Figure 8. Firstly, 3% Au/P-g-C$_3$N$_4$ had enhanced light utilization, which resulted from the P doping narrowing the band gap of g-C$_3$N$_4$ and an intense absorption in the region of

450–700 nm induced by SPR of Au nanoparticles. Secondly, an intense near-electric field induced by this SPR improved the efficiency of carrier separation. Meanwhile, when Au nanoparticles combined with P-g-$C_3N_4$, due to the difference of the work function, the free electrons would transfer from P-g-$C_3N_4$ to Au. The Fermi levels and the band position would change at interface. Finally, a Schottky barrier formed to hinder the electron transfer from gold to P-g-$C_3N_4$. Under light illumination, P-g-$C_3N_4$ could be excited to generate holes and electrons, the excited electrons from the CB of P-g-$C_3N_4$ transferred to the Au, resulting in a high efficiency of carrier separation and transfer [31]. Then the electrons would react with $H^+$ for the generation of $H_2$ and the holes would be consumed by sacrificial agents. Hence, the 3% Au/P-g-$C_3N_4$ photocatalyst showed excellent photocatalytic activities.

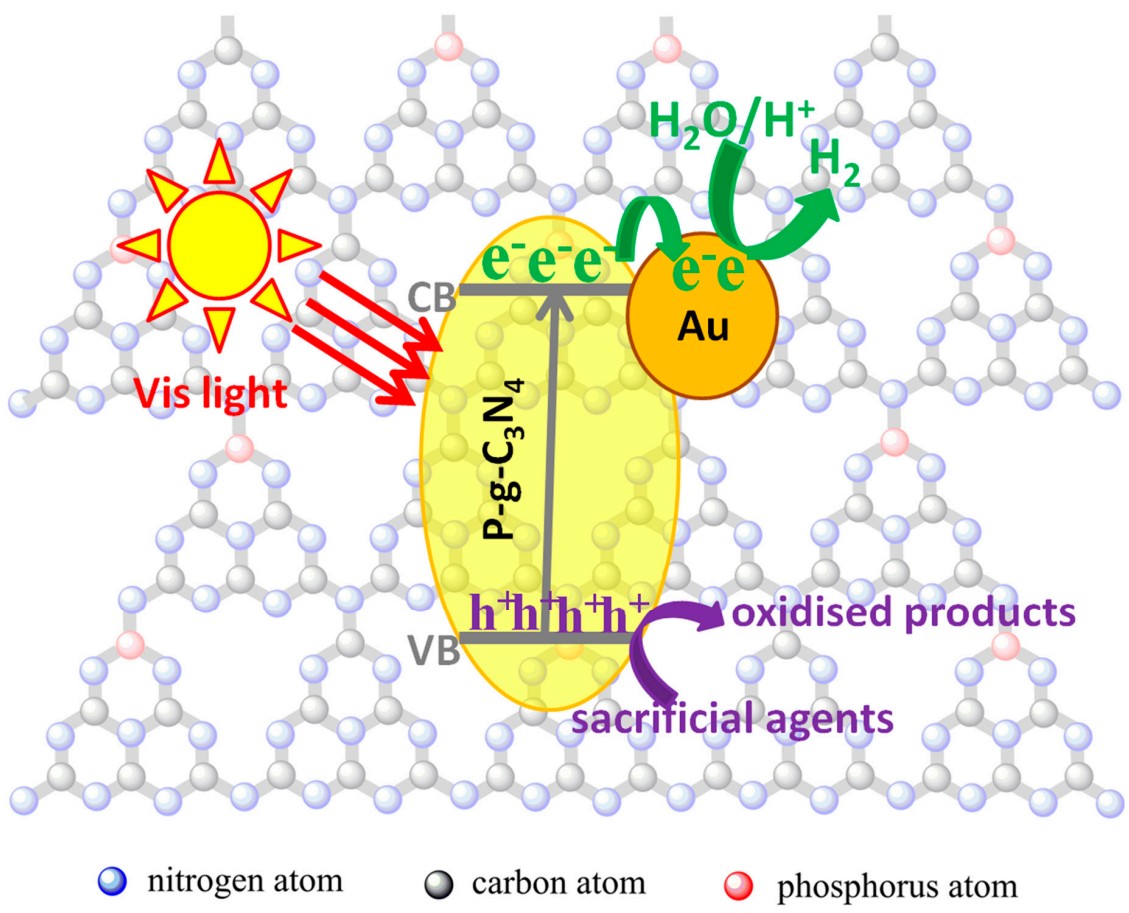

**Figure 8.** The mechanical illustration of enhanced photocatalytic activity for 3% Au/P-g-$C_3N_4$ sample.

## 3. Materials and Methods

### 3.1. Synthesis of g-$C_3N_4$ and P-g-$C_3N_4$

The g-$C_3N_4$ was prepared by polymerization of urea, 7 g urea was thermally treated in a tube furnace at 550 °C for 4 h with a heating rate of 10 °C min$^{-1}$. When the ceramic container was cooled to room temperature, the yellow-colored product (g-$C_3N_4$) was collected and ground into powder.

The P doped g-$C_3N_4$ was prepared by copolymerization of the precursor according to the literature [1]. In a typical process, 7 g urea and 0.1 g $NH_4H_2PO_4$ were ground together in an agate mortar and then thermally treated in a tube furnace at 550 °C for 4 h with a heating rate of 10 °C min$^{-1}$. When the ceramic container was cooled to room temperature, the yellow-colored product (P-g-$C_3N_4$) was collected and ground into powder.

### 3.2. Synthesis of Composites

In a typical preparation of 3% Au/P-g-C$_3$N$_4$, 150 mg of P-g-C$_3$N$_4$ was dispersed in water with mild sonication (10 min). A 3 mL sample of isopropanol and 7.8 mL HAuCl$_4$ were added to the above suspension then with sonication (2 h, dark). The resulting mixture was stirred under UV light illumination (1.5 h) with continuous nitrogen sparging. Then the mixture was washed thoroughly with distilled water and finally dried in a vacuum oven at 50 °C. The nominal Au contents were 1, 3, and 5 wt.% for the Au/P-g-C$_3$N$_4$ composites using different amounts of HAuCl$_4$. Au/g-C$_3$N$_4$ was synthesized under the same conditions using the crude g-C$_3$N$_4$ synthesized as above.

### 3.3. Characterizations

The crystal structure of the samples was tested by X ray powder diffraction (XRD), the parameters of XRD were graphite monochromator, $\lambda$ = 1.54184Å, room temperature, Bruker D2 PHASER X ray diffractometer. The valence states and chemical composition of samples was tested by X-ray photoelectron spectroscopy (XPS), the parameters of XPS were a monochromated Al K$\alpha$ irradiation, PHI-5702, Physical Electronics. High-resolution transmission electron microscopy and transmission electron microscopy (HRTEM, TEM, Tecnai F30, 300 kV operated voltage) were used to observe the morphologies of the samples. The specific surface area (BET) of the sample was measured by a Micromeritics ASAP 2000 system. Perkin Elmer 950 spectrometer (DRS) and Photoluminescence emission spectra (PL, FLS-920T, 450 W xenon arc lamp) were used to test the optics and electronic properties. The type of electrochemical workstation was a CS 310, Wuhan Corrtest Instrument Co. Ltd. Reference electrodes (an Ag/AgCl electrode and a platinum foil electrode) with working electrodes were immersed in 0.1 M Na$_2$SO$_4$ aqueous solution (300 W Xe lamp irradiation) for the photoelectrochemical properties tested. All photoelectrochemical measurements were carried out at a constant electrode potential of 0 V to the reference electrode. In a typical preparation of the electrode, 15 mg 3% Au/P-g-C$_3$N$_4$, 40 μL terpilenol, 10 μL alcohol and 10 μL Nafion aq.(5 wt.%) were ground together, then the composite was coated on FTO (effective area is cm$^2$) as a working electrode.

### 3.4. Evaluation of Photocatalytic Activity

An automatic gas circulation system (CEL-SPH2N-D9) was used for H$_2$ generation experiments. Methanol was employed as the sacrificial agent. In a typical process, 50 mg of sample was dispersed in a mixed solution (40 mL H$_2$O and 10 mL methanol). After sonication, the reaction unit and gas circulation system were thoroughly degassed for 0.5 h with a stationary temperature at 6 °C using a cooling device. Then the mixed solution was irradiated under a 300 W Xe lamp. Visible light (420–780 nm) was obtained using a UV-IR-CUT filter. Finally, an on-line gas chromatograph (GC-7920) equipped with a thermal conductivity detector (TCD) was employed for data acquisition, which used carrier gas (N$_2$) and 5Å molecular sieve column.

## 4. Conclusions

In summary, Au/P-g-C$_3$N$_4$ photocatalysts were prepared by facile thermal copolymerization and in situ photodeposition methods. The P element was doped into the structure of g-C$_3$N$_4$ marginally and the monodispersed Au nanoparticles were adequately anchored on the surface of P-g-C$_3$N$_4$. The obtained Au/P-g-C$_3$N$_4$ photocatalysts had the enhanced visible light utilization, improved photogenerated electron-hole pairs separation efficiency as well as abundant reactive sites, which were derived from the synergic effect between Au caused by the surface plasmon resonance effect and the dopant phosphorus-induced structural and electronic properties changing. Hence, Au/P-g-C$_3$N$_4$ photocatalysts show arresting photocatalytic activity under visible light illumination. This work could provide a promising approach to develop high-efficiency photocatalysts applied to H$_2$ generation.

**Supplementary Materials:** The following are available online at http://www.mdpi.com/2073-4344/10/6/701/s1, Figure S1: TEM images of g-C$_3$N$_4$ (a), P-g-C$_3$N$_4$ (b), 3% Au/g-C$_3$N$_4$ (c), 1% Au/P-g-C$_3$N$_4$ (d), 3% Au/P-g-C$_3$N$_4$ (e) and 5% Au/P-g-C$_3$N$_4$ (f), Figure S2: N$_2$ adsorption-desorption isotherms of g-C$_3$N$_4$, P-g-C$_3$N$_4$ and 3% Au/P-g-C$_3$N$_4$, Figure S3: XRD patterns of fresh and used 3% Au/P-g-C$_3$N$_4$, Figure S4: XPS spectrum and TEM photograph of used 3% Au/P-g-C$_3$N$_4$.

**Author Contributions:** Data curation, H.L.; Formal analysis, H.L.; Funding acquisition, Y.W.; Project administration, Y.W.; Writing—original draft, H.L.; Writing—review & editing, N.Z., F.Z., T.L. and Y.W. All authors have read and agreed to the published version of the manuscript.

**Funding:** This work is supported by the National Natural Science Funds of China (Nos. 51672115 and 21501080) and the Gansu Province Development and Reform Commission (NDRC, No. 2013-1336).

**Conflicts of Interest:** The authors declare no conflict of interest.

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
