# Peer review of "Facile Fabrication of a Novel Au/Phosphorus-Doped g-C3N4 Photocatalyst with Excellent Visible Light Photocatalytic Activity"

_catalysts, doi:10.3390/catal10060701_

Round 1
Reviewer 1 Report
The authors have replied to the majority of the comments, therefore the paper can be accepted.
Reviewer 2 Report
The manuscript is beginning to take a good shape now. However there remain a large number of grammatical errors.
The abstract is concise but still needs attention to make complete sense.
For the highlighted sentence (lines 16,17) I would write, logically the following...'The results showed that phosphorus was successfully doped into the structure of gC3N4 and that surface deposition of gold was successfully accomplished' You can then delete the next sentence (18,19) because it says exactly the same !
Line 20 it should be 'compared'
Line 20-23 might better read 'The enhancement of photocatalytic activity is due to the synergic effect between gold-induced surface plasmon resonance and the modified structural and electronic properties of the gC3N4 induced by the phosphorus dopant' - it now makes sense.
There are grammatical errors in line 26 (x2), 36 (x2), 55-57, but in general, the introduction is reading much better - well done.
Line 57 you could put 'conjugated aromatic Π-stacking'
Line 72 'units'
Line 80 'corresponding' not 'which indicating'
Line 87 'which shows that the introduction of Au particles ….'
Line 98,99 needs to make more sense
115 during the doping process
119 delete 'and'
163 compared
165 delete 'all'
166-167 grammar errors
179 should read 'was maintained with minimal leaching of P and Au.'
Figure 8 - You must still do something with the figure, because the reader can see nothing. I suggest making the poly-melem structure very feint, much lighter on the grey scale, then we will be able to see the important stuff. We cannot see the important bits clearly at the moment.
Materials and methods are much clearer (line 218 'thermally' not thermal).
line 233 'under the same conditions using the crude g-C3N4 synthesised as above.'
Tidy the manuscript as suggested, work on improving the diagram and I'll take one last look...
Thank you for the improvements you have made.
Reviewer 3 Report
Owing to the appropriate revision by the authors, the manuscript has been significantly improved. Although some minor revisions are still needed, I almost consider that this manuscript can be recommended for publication.
- What is the electrode potential used for the photoelectrochemical experiments?
- How did the author prepare the electrode for the photoelectrochemical experiments? The detail of the preparation method should be provided.
Author Response
Please see the attachment.

This manuscript is a resubmission of an earlier submission. The following is a list of the peer review reports and author responses from that submission.
Round 1
Reviewer 1 Report
- The authors should present in the materials and methods the synthesis of C3N4.
2. For the XPS data they could add the % Au and %P coverage on the surface and make comparison with the catalytic trend.
3. Tables with TEM and XPS, UV data should be added (mean particle size, particle size distribution, BE values and percentage of species, band gap, etc). The discussion based on reaction rates for the catalytic trend would be useful including the data from characterisation.
4. TEM images for the whole series of catalysts should be added including histograms.
5. For the photocatalytic stability the authors should present characterisation data of the used catalyst and also check for leaching of P and Au in the solution.
6. The quality of the figures is poor and the authors should check carefully, in some cases it is difficult to read.
7. In general, the presentaion of the manuscript could significanlty improved.
8. The references added are few. They should consider incireasing the number and checking recent papers and reviews in this field.
I recomend major revision.
Reviewer 2 Report
Firstly, I would like to say that I see merit in this work from looking at the diagrams. The PCA in Figure 5 is very promising, and the cycling runs (note not circling) good potential.
Unfortunately, at this stage however, I cannot provide a detailed review because the manuscript is very difficult to read, and extracting the scientific detail is not easy for the reader.
I recommend a revision based around a re-writing of the text throughout.
I look forward to reviewing the revised and rewritten manuscript, these materials seem promising from the diagrams.
A couple of pointers to start,
1) sentence 1 of the introduction needs rewriting
2) Line 32 you should introduce the material thus... 'Graphitic carbon nitride, g-C3N4, is a non-metal polymer semiconductor that has shown much promise as a visible light photocatalyst for a range of applications from hydrogen (H2) generation to organic pollutant digestion [references needed here please].
3) A minor point, but important is that the chemical symbols throughout the text, should have subscripts, as I have written in (2)not small letters on the same line.
4) I find the references are confusing as written, please put the year, followed by the volume (in bold), then the page numbers. Please make them consistent throughout, without random periods as you have written in 4, 5 and 6. As an example the detail of reference 1 could be
2017, 5, 5831-5841 or 5, 5831-5841 (2017)
make sure it is consistent throughout, whichever way you do it.
5) The critical summary picture, Figure 8, is not at all clear from either the diagram or the explanation (lines 174-178).
Reviewer 3 Report
In this paper, the authors synthesized Au / P-doped C3N4 as a visible light-sensitive photocatalyst for H2 evolution. At present, solar H2 production using photocatalyst is a very important research topic, but the quality of this manuscript has not reached an acceptable level for publication. Reviewer recommends to resubmit the manuscripts after fully addressing the following concerns:
- In Figure 4, visible light sensitivity is almost the same between C3N4 and P-doped C3N4. On the other hand, P-doping highly improved visible light utilization ability in ref. 11. So, I consider that the content of P doped in this study is quite low and insufficient to obtain visible light sensitivity. The author should quantify the amount of P doped in C3N4.
- What is the surface areas of all the sample subjected to the photocatalytic experiments?
- What reaction does photocurrent originate from? What is the electrode potential used for the photoelectrochemical experiment?
- Why did 1% Au/P-C3N4 show stronger absorption than 3%- and 5%-Au/P-C3N4 in the wavelength region between 500 and 700 nm?
Round 2
Reviewer 1 Report
The manuscript has improved and the authors have answered the majority of the comments. However, for the used catalyst should provide, XPS, TEM and ICP analysis since P and Au can leach in the solution and XRD is not a precised technique for characterising small nanoparticles. It is a good indication but not enough.
I recommend revision and acceptance of the paper later if the authors answer to the comments.
Reviewer 2 Report
While some technical aspects of the paper have improved, there is still much work on the quality of the manuscript to be done before it can be approved for publication. A lot of the comments relate to your attention to detail and making sure the chemcial formulae are consistent throughout. The format also changes. You must strive for consistency throughout.
- The first two paragraphs of the introduction need to be clearer. You made some godd changes, but you have also made some not so good ones, it remains difficult for the reader.
- Somewhere around line 72 it would be good to show the structure of melem, then everyone will have a good idea of what the whole paper is about. maybe you should give a chemical name here too ( 1,3,4,6,7,9,9b-Heptaazaphenalene-2,5,8-triamine) this would make more sense.
- Also in the lines 72 to 88 there is either a spelling mistake, a grammatical error or a typographical error on every line. You must attend to this before approval
- The sentence on line 92-93 remains a mystery to me
- line 113 to 125 are a bit similar to point 3, please rewrite to make sense with no errors
- lines 126-130 you need subscripts in the formulae for C3N4
- lines 159 - 172 again have many errors.and also 182 to 190
- in 196 normally we do not say violent when describing absorption.
- Figure 8 still needs work to make it clearer, it is difficult for the reader to find everything
- Materials and Methods
- paragraph 1 how did you isolate or 'obtain' the sample, you must describe as you have done in lines 213 to 219, where it is clear
- line 209, ground together, not grounded.
- Again, in general you did not heed my worries about formatting the chemical formulae. I give you an example.... if I write the formula, I get g-C3N4 and you see these are subscript. If you write the same thing you put g-C3N4 just with small numbers (I cannot do it exactly here). We need them to be subscripts throughout.
- I think you have done a good job on the references.
In summary, as I said before there is some good underlying science here, but the presentation is not appropriate for publication in 'Catalysts' at the moment. Put some work in to 'attention to detail', consistency and clarity of phrases. Improve the materials and methods.
Good luck, time doesn't matter but exellence does. I look forward to seeing this manuscript again.
Reviewer 3 Report
The authors revised the manuscript based on the reviewers' comments, and its quality has improved. However, I consider that the manuscript needs to be further improved for publication.
- In abstract and conclusion, the authors described that doping of phosphorus into g-C3N4 was accomplished successfully. But, as shown in the revised manuscript, the amount of doped P is quite low. So, I consider that the synthesis of Au/P-C3N4 was not successful at least about the doping of phosphorus. I strongly recommend to modify the description relevant to it.
- In the reply to my comment 3, the author hasn't answered the origin of the photocurrent. I know that photoexcited electrons or photogenerated holes generate photocurrent. I'm asking which one contributed to the photocurrent generation in this case. And I'm also asking what kind of reaction occurs on the surface of photoelectrode during photocurrent generation.